# HEPA: A Self-Supervised Horizon-Conditioned Event Predictive Architecture for Time Series

**Jonas Petersen** [* 1 2]  **Alessandro Lombardi** [2]  **Riccardo Maggioni** [2]
**Camilla Mazzoleni** [2]  **Federico Martelli** [1 2]  **Philipp Petersen** [3]

## Abstract

Critical events in multivariate time series, from turbine failures to cardiac arrhythmias, demand accurate prediction, yet labeled data is scarce because such events are rare and costly to annotate. We introduce HEPA (Horizon-conditioned Event Predictive Architecture), built on two key principles. First, a causal Transformer encoder is pretrained via a Joint-Embedding Predictive Architecture (JEPA): a horizon-conditioned predictor learns to forecast future *representations* rather than future values, forcing the encoder to capture predictable temporal dynamics from unlabeled data alone. Second, we freeze the encoder and *finetune only the predictor* toward the target event, producing a monotonic survival cumulative distribution function (CDF) over horizons. With fixed architecture and optimiser hyperparameters across all benchmarks, HEPA handles water contamination, cyberattack detection, volatility regimes, and eight further event types across 11 domains, exceeding leading time-series architectures including PatchTST, iTransformer, MAE, and Chronos-2 on at least 10 of 14 benchmarks, with an order of magnitude fewer tuned parameters and, on lifecycle datasets, an order of magnitude less labeled data.

## 1. Introduction

A turbine blade cracks after 12,000 flight hours. A satellite sensor drifts silently for 48 hours before triggering a cascade. A cardiac arrhythmia emerges from subtle ECG deviations. These events are rare, yet they follow partially predictable precursor dynamics (Scheffer et al., 2009). Remaining-useful-life models (Fan et al., 2024) estimate time-to-failure; anomaly detectors (Xu et al., 2022; He et al., 2026) flag abnormal readings. Yet all these tasks share one structure: given observations up to time $t$, estimate $P(\text{event within } \Delta t)$ for each prediction horizon $\Delta t$.

This structural uniformity suggests a separation of concerns. The *encoder* learns temporal dynamics from unlabeled data; the *predictor*, finetuned with a small number of event labels, specialises to whichever event is relevant. The key design choice is what the encoder should forecast during pretraining. Value-forecasting approaches, whether supervised (Nie et al., 2023) or pretrained on large corpora (Ansari et al., 2025; Das et al., 2024; Goswami et al., 2024), shape representations around all variation in the signal, including noise irrelevant to the downstream event. The Joint-Embedding Predictive Architecture (JEPA) (Assran et al., 2023) offers an alternative: by forecasting future *representations* rather than future values, the encoder retains what is predictable about the future and discards what is not.

We apply this principle to time series as HEPA. A causal Transformer encodes observations up to time $t$; a horizon-conditioned predictor maps the encoding and a horizon $\Delta t$ to a predicted future representation, forcing the encoder to internalise dynamics at multiple timescales. After self-supervised pretraining, we retain the predictor: freeze the encoder but finetune the predictor alongside a lightweight event head that outputs a discrete-time survival CDF, ensuring the predicted event probability never decreases as the horizon grows. This "predictor finetuning" tunes only 198K parameters ($11\times$ fewer than end-to-end training) yet is more expressive than a linear probe because the predictor reshapes its horizon-conditioned outputs to align with the downstream event (Section D).

Our contributions:

1. **One architecture, any event, any domain.** A single 2.16 M-parameter architecture with fixed hyperparameters, evaluated on 14 benchmarks across 11 domains. HEPA wins on 10 of 14 while tuning $11\times$ fewer parameters. What transfers across domains is the *recipe*

---

[*]Equal contribution  [1]ETH Zürich [2]Forgis [3]University of Vienna. Correspondence to: Jonas Petersen <jonas.petersen@forgis.com>.

*Proceedings of the $2^{nd}$ ICML Workshop on Foundation Models for Structured Data*, Seoul, South Korea. 2026. Copyright 2026 by the author(s).

(architecture + JEPA pretraining + predictor finetuning), not the weights.

2. **Predictor finetuning as downstream recipe.** On C-MAPSS (Saxena et al., 2008), HEPA retains 92% of full-label h-AUROC at just 2% of labels. An information-theoretic bound (Section F) formalises when and why this works.

An extended version with full ablations, per-seed results, and additional baselines (MOMENT, TFM-2.5, Moirai, MTS-JEPA) is available at the link above.

## 2. Related Work

**Self-supervised learning for time series.** SSL for time-series representation learning falls into three families. Contrastive methods, including TS2Vec (Yue et al., 2022), TNC (Tonekaboni et al., 2021), and CPC (van den Oord et al., 2018), learn representations by contrasting positive and negative pairs. Masked reconstruction approaches such as PatchTST (Nie et al., 2023) and SimMTM (Dong et al., 2023) recover masked patches in input space. JEPA (Assran et al., 2023; Bardes et al., 2024) predicts future *representations* rather than reconstructing inputs, avoiding tying the latent space to value-level fidelity. For time series, TS-JEPA (Ennadir et al., 2024) applies temporal masking for classification and MTS-JEPA (He et al., 2026) adds code-book regularisation for anomaly detection. All these methods discard their pretraining head at inference and probe only the encoder. HEPA instead retains the predictor and finetunes it toward the downstream event.

**Foundation models for time series.** Chronos-2 (Ansari et al., 2025), TFM-2.5 (Das et al., 2024), MO-MENT (Goswami et al., 2024), and Moirai (Woo et al., 2024) pretrain on large-scale corpora for generic value forecasting. These approaches target future channel values; HEPA targets event probabilities. The encoder is mid-scale and pretrained per-dataset; what transfers across domains is the *recipe*, not the weights. We benchmark HEPA against four of these foundation models using identical downstream heads to isolate encoder quality.

**Prognostics and survival modelling.** HEPA's downstream parameterisation builds on discrete-time survival models (Lee et al., 2018; Gensheimer & Narasimhan, 2019), which decompose event probability into per-interval hazards composed into a survival CDF. We adapt this to a multi-horizon event prediction setting and evaluate via h-AUROC, unifying RUL estimation and anomaly detection under one metric. Related event-forecasting approaches include Time-Cast (Nakamura et al., 2026), which mines time-evolving stages in multi-sensor streams and fits a per-stage model for online time-to-event prediction, and EVEREST (Zilin-skas et al., 2026), which uses auxiliary heads for rare-event calibration. HEPA differs from both by requiring no stage structure or domain-specific auxiliary losses; the survival CDF emerges from a generic JEPA recipe.

## 3. Method

**Architecture and pretraining.** HEPA consists of three components (Figure 1). The **context encoder** $f_\theta$ is a causal Transformer ($d$=256, 2 layers, 4 heads) that maps observations $\mathbf{x}_{\leq t}$, tokenised into patches of size $P$=16 with instance normalisation (Kim et al., 2022b) and sinusoidal positional encodings, to $\mathbf{h}_t \in \mathbb{R}^d$. The **predictor** $g_\phi$ is a 2-layer MLP that takes $\mathbf{h}_t$ and a horizon $\Delta t$ (sampled log-uniformly during pretraining) to produce $\hat{\mathbf{h}}_{(t,t+\Delta t]}$. The **target encoder** $\bar{f}_\theta$ is a weight-shared copy of $f_\theta$ applied bidirectionally with attention pooling; both encoders are trained jointly. A SIGReg regulariser (Balestriero & LeCun, 2025) prevents collapse by constraining predicted representations toward an isotropic Gaussian, replacing the EMA schedule of standard JEPA. The pretraining loss is:

$$\mathcal{L} = (1 - \alpha)\|\hat{\mathbf{h}} - \mathbf{h}^*\|_1 + \alpha\,\mathcal{L}_{\text{SIG}}, \tag{1}$$

where $\alpha$=0.1. No labels are used. Pretraining takes under one minute per dataset on a single A10G GPU.

**Predictor finetuning.** After pretraining, we freeze $f_\theta$ and finetune only $g_\phi$ plus a linear event head (198K params total). The predictor runs at $K$ unit-step horizons ($K$=150 for C-MAPSS/TEP, $K$=200 otherwise). Each predicted representation maps to a per-interval conditional hazard $\lambda_{\Delta t}(t) = \sigma(\mathbf{w}^\top \hat{\mathbf{h}}_{(t,t+\Delta t]} + b) \in (0, 1)$. The event probability surface is a discrete-time survival CDF (Lee et al., 2018; Gensheimer & Narasimhan, 2019):

$$p(t, \Delta t) = 1 - \prod_{j=1}^{\Delta t}(1 - \lambda_j(t)). \tag{2}$$

Because each factor $(1 - \lambda_j) \in (0, 1)$, monotonicity in $\Delta t$ holds by construction. The finetuning loss sums positive-weighted BCE over horizons:

$$\mathcal{L}_{\text{FT}} = \sum_{\Delta t=1}^{K} w^+ \cdot \text{BCE}\big(p(t, \Delta t),\ y(t, \Delta t)\big), \tag{3}$$

where $y(t, \Delta t) = \mathbb{1}[\text{event in } (t, t+\Delta t]]$ and $w^+ = N_{\text{neg}}/N_{\text{pos}}$.

**Why does predictor finetuning work?** An information-theoretic bound (Section F) formalises when and why the pretrained encoder retains enough event-relevant information for a small downstream head to extract it. The bound predicts that lower pretraining loss $\varepsilon$ implies higher mutual information between the encoder output and the event,

**Stage 1: Self-Supervised Pretraining**

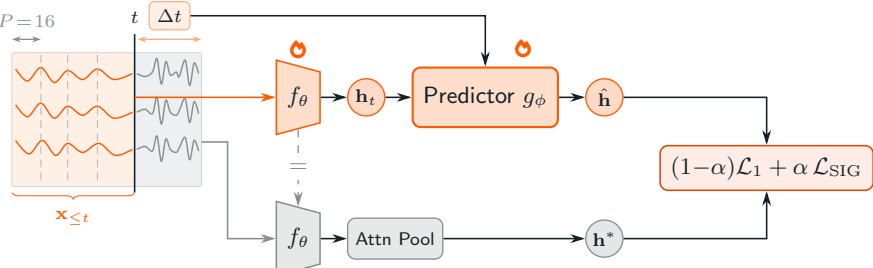

**Stage 2: Supervised Finetuning**

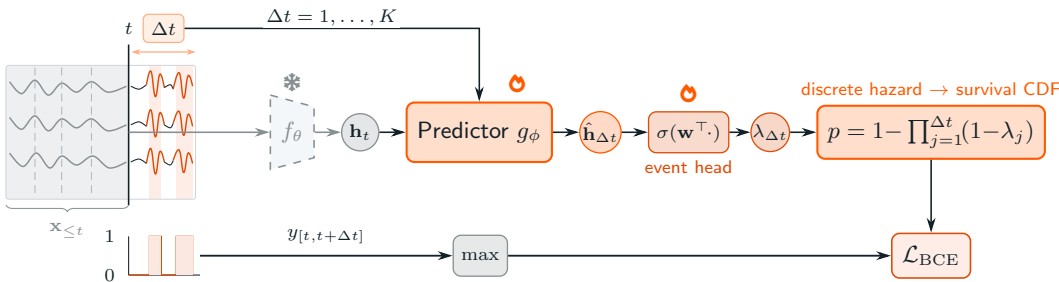

*Figure 1.* **HEPA architecture.** Both stages sweep over all $(t, \Delta t)$ pairs per episode. *Stage 1 (self-supervised pretraining):* The causal encoder $f_\theta$ maps observations $\mathbf{x}_{\leq t}$ to $\mathbf{h}_t$; the predictor $g_\phi(\mathbf{h}_t, \Delta t)$ predicts future representations via a JEPA objective; the target encoder produces $\mathbf{h}^*$ from the future interval. SIGReg prevents collapse. *Stage 2 (supervised finetuning):* Encoder frozen; the predictor produces $K$ horizon-specific hazard rates $\lambda_{\Delta t}$ composed into a monotonic survival CDF $p(t, \Delta t)$. Only the predictor and event head (198K of 2.16M parameters) are finetuned.

which should manifest as stronger downstream h-AUROC. We validate this within three datasets by snapshotting the encoder during pretraining at epochs $\{1, 3, 8, 25\}$ plus the converged best, and running predictor finetuning at each snapshot (3 seeds). The monotone prediction holds: Spearman $\rho(\varepsilon, \text{h-AUROC}) = -0.67$ ($p{=}0.017$) on C-MAPSS-3, $\rho{=}{-}0.64$ ($p{=}0.026$) on MBA, and $\rho{=}{-}0.49$ ($p{=}0.13$) on SMAP (Figure 2 in Section F). The bound also predicts when the recipe *fails*: on datasets without extended precursors (e.g. GECCO), we observe $\rho{=}{+}0.14$ ($p{=}0.67$), exactly as expected when the bound is vacuous.

## 4. Experiments

**Setup.** We pretrain a separate HEPA encoder per dataset from unlabeled training data. Architecture and hyperparameters are identical across all domains; only the input projection (sensor count $S$) changes. All comparison methods share the same downstream MLP head, positive-weighted BCE loss, and evaluation protocol; only the frozen encoder differs (Section C). The dataset overview (14 datasets, 11 domains) is in Section A.

**Evaluation metric.** The model outputs a probability surface $p(t, \Delta t)$ for each observation time $t$ and prediction horizon $\Delta t$. This surface is the complete prediction; every metric is computed deterministically from it. As a cross-domain metric, we use **h-AUROC**: the mean of per-horizon AUROC values computed over $p(t, \Delta t)$. Per-horizon prevalence varies wildly even within a single surface: on C-MAPSS-1, the event "failure within $\Delta t$ steps" has prevalence 0.5% at $\Delta t{=}1$ and 96% at $\Delta t{=}150$, a $\sim200\times$ range. Pooled AUPRC over all $(t, \Delta t)$ cells inherits a 0.957 baseline on C-MAPSS-1, because a model predicting only per-horizon prevalence already scores there. h-AUROC avoids this by decomposing the surface into independent per-horizon binary classification problems, each with a universal 0.5 baseline that does not depend on prevalence. Domain-specific metrics (RMSE for remaining-useful-life, PA-F1 for anomaly detection) are derived as projections of the same surface for comparability with published baselines. All numbers are mean $\pm$ std over 5 seeds (HEPA, PatchTST, iTransformer, MAE) or 3 seeds (Chronos-2).

**Main results.** Table 1 compares HEPA against PatchTST (Nie et al., 2023), iTransformer (Liu et al., 2024),

*Table 1.* **Main results** (mean ± std; 5 seeds for HEPA, PatchTST, iTransformer, MAE; 3 seeds for Chronos-2). All methods use matched-capacity downstream heads on frozen encoders. **Bold** = best mean per row. Domain-specific metrics (RMSE, PA-F1, F1) derived from the same surface.

| Dataset | Domain | Label % | FM Chr-2 | PatchTST | iTransf h-AUROC ↑ | MAE | **HEPA** | Metric | **HEPA** domain metric | SOTA |
|---|---|---|---|---|---|---|---|---|---|---|
| C-MAPSS-1 turbine failure | Turbo. | 100 | .66±.00 | .80±.04 | .70±.05 | .69±.02 | **.81**±.03 | RMSE↓ | 28.5 | **12.2** |
| | | 10 | .66±.00 | .69±.08 | .59±.04 | .63±.12 | **.78**±.07 | | 32.3 | **17.0** |
| C-MAPSS-2 multi-cond. | Turbo. | 100 | .45±.01 | .44±.03 | .43±.03 | .56±.01 | **.57**±.01 | RMSE↓ | 40.8 | **20.0** |
| | | 10 | .46±.02 | .48±.08 | .50±.08 | .52±.01 | **.55**±.01 | | **40.9** | 43.1 |
| C-MAPSS-3 multi-fault | Turbo. | 100 | .73±.00 | .79±.01 | .76±.01 | .78±.02 | **.84**±.01 | RMSE↓ | 34.7 | **12.7** |
| | | 10 | .67±.00 | .77±.03 | .64±.06 | .78±.03 | **.84**±.01 | | 47.1 | **21.8** |
| C-MAPSS-4 multi-cond.+fault | Turbo. | 100 | — | .52±.03 | .45±.02 | .57±.02 | **.63**±.02 | | | |
| | | 10 | — | .51±.04 | .49±.04 | .52±.04 | **.55**±.03 | | | |
| SMAP sensor anomaly | Spacecraft | 100 | .54±.02 | .49±.03 | .47±.05 | **.64**±.04 | .59±.05 | PA-F1↑ | .94 | **.96** |
| | | 10 | .51±.01 | .46±.05 | **.52**±.07 | .50±.15 | .49±.08 | | .92 | **.96** |
| PSM server anomaly | Server | 100 | .48±.01 | .55±.02 | .52±.02 | .56±.02 | **.57**±.02 | PA-F1↑ | .94 | **.98** |
| | | 10 | .51±.01 | .43±.03 | **.53**±.01 | .52±.02 | .48±.04 | | .95 | **.98** |
| MBA arrhythmia | Cardiac | 100 | .53±.10 | .68±.07 | **.84**±.03 | .73±.03 | .75±.03 | F1↑ | **.98** | .77 |
| | | 10 | .50±.07 | **.65**±.08 | .31±.05 | .53±.04 | .55±.15 | | **.98** | .81 |
| BATADAL cyberattack | ICS | 100 | .56±.01 | **.66**±.03 | .46±.13 | .60±.04 | .57±.03 | F1↑ | **.77** | .74 |
| | | 10 | .55±.04 | .37±.04 | .54±.08 | **.61**±.08 | .54±.06 | | **.61** | .33 |
| TEP process fault | Chemical | 100 | — | .99±.02 | .93±.02 | .96±.02 | **1.00**±.00 | F1↑ | **.95** | .93 |
| | | 10 | — | **1.00**±.00 | .94±.05 | .99±.00 | **1.00**±.00 | | .88±.03 | .86 |
| ETTm1 overheating | Power | 100 | .74±.01 | .53±.03 | .79±.03 | **.87**±.00 | .81±.00 | | | |
| | | 10 | .68±.02 | .50±.03 | .61±.02 | **.77**±.02 | .73±.01 | | | |
| Weather heat spike | Climate | 100 | .72±.02 | .64±.08 | .83±.04 | .88±.02 | **.89**±.01 | | | |
| | | 10 | .71±.02 | .67±.03 | **.85**±.01 | .83±.02 | .83±.02 | | | |
| Beijing-AQ PM2.5 spike | Air | 100 | .53±.00 | **.81**±.01[†] | .75±.02 | **.81**±.01 | **.81**±.01 | | | |
| | | 10 | .49±.00 | .65±.08 | .73±.02 | .77±.01 | **.78**±.02 | | | |
| VIX vol regime | Finance | 100 | .40±.04 | .48±.11 | .40±.14 | **.57**±.03 | **.57**±.01 | | | |
| | | 10 | .38±.04 | **.57**±.13 | .48±.05 | .55±.03 | .55±.01 | | | |
| GECCO contamination | Water | 100 | .74±.02 | .65±.07 | .64±.15 | .81±.04 | **.88**±.06 | | | |
| | | 10 | **.80**±.03 | .52±.11 | .50±.13 | .55±.11 | .39±.13 | | | |
| *Best (100%)* | | | 0 | 2 | 1 | 4 | 10 | | | |

Matched downstream heads (HEPA 198K pred-FT; baselines 264K dt-MLP), positive-weighted BCE, identical protocol. $K=150$ for C-MAPSS/TEP, $K=200$ otherwise. PA-F1 = point-adjusted F1 (Kim et al., 2022a). [†]Beijing-AQ PatchTST: 3 stations at 100%. Additional baselines (MOMENT, TFM-2.5, Morai, MTS-JEPA) in extended version.

MAE (same architecture, reconstruction objective), and Chronos-2 (Ansari et al., 2025) (119M-param foundation model). HEPA wins on 10 of 14 benchmarks at 100% labels, including all four C-MAPSS variants and TEP (perfect h-AUROC). JEPA pretraining captures temporal structure that supervised and reconstruction-based methods miss on datasets with extended precursor dynamics (C-MAPSS, GECCO, PSM). MAE wins on SMAP and ETTm1 (gradual-drift failures); iTransformer wins on MBA (sensor-localised arrhythmia). HEPA matches or exceeds Chronos-2 despite $55\times$ fewer parameters and no external data; all foundation models are evaluated as frozen encoders (end-to-end adaptation could improve their standing). Additional comparisons: MTS-JEPA (He et al., 2026) (Section G), MOMENT, TFM-2.5, Morai (Section E).

**Label efficiency and failure modes.** Without labels, the encoder learns a degradation manifold (PC1 captures 61% of variance; Section F), so HEPA retains 92% of full-label h-AUROC with just 2% of labels on C-MAPSS-1 (Section B). HEPA loses on four datasets, traceable to channel-fusion tokenisation diluting localised signals (BATADAL, MBA) or MAE reconstruction transferring better for gradual drift (SMAP, ETTm1); sensor-as-token strategies (Liu et al.,

2024) are the natural fix.

## 5. Conclusion

HEPA demonstrates that self-supervised JEPA pretraining combined with predictor finetuning provides a practical, domain-agnostic recipe for event prediction. The encoder learns temporal dynamics from unlabelled data; the predictor learns which dynamics signal the target event. One architecture handles 14 benchmarks in 11 domains, matching or exceeding PatchTST, iTransformer, MAE, and Chronos-2 on the majority while tuning $11\times$ fewer parameters, and retaining 92% of full-label performance with 2% of labels on lifecycle datasets (Section F). Current limitations are CDF-level BCE distorting calibration (post-hoc Platt scaling recommended) and channel-fusion tokenisation failing on sensor-localised events. Cross-domain pretraining and sensor-as-token strategies (Liu et al., 2024) are natural next steps toward a foundation model whose weights, not just recipe, transfer across domains.

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

*Table 2.* Dataset overview (14 benchmarks, 11 domains). Patch size $P=16$, context 512 steps except C-MAPSS (full engine history).

| Dataset | Target event | Sensors | Rate |
|---|---|---|---|
| C-MAPSS FD001–FD004 | Engine failure | 14 | 1/cycle |
| SMAP | Spacecraft fault | 25 | 1 Hz |
| PSM | Server fault | 25 | 1/min |
| MBA (ECG) | Cardiac arrhythmia | 2 | 275 Hz |
| GECCO (water) | Water contamination | 9 | 1/min |
| ETTm1 | Transformer overheating | 7 | 15/min |
| BATADAL (ICS) | Cyber-attack on SCADA | 43 | 1/hour |
| TEP | Process fault | 52 | 1/3 min |
| Weather | Heat spike | 21 | 10/min |
| Beijing-AQ | PM2.5 spike | 11 | 1/hour |
| VIX | Volatility regime | 6 | 1/day |

*Table 3.* **Label efficiency on C-MAPSS (HEPA, 3 seeds, h-AUROC).** Retention relative to full labels. C-MAPSS-1 retains 92% at 2% labels (2 of 85 engines).

| | C-MAPSS-1 (85 eng.) | | | C-MAPSS-3 (100 eng.) | | |
|---|---|---|---|---|---|---|
| Labels | h-AUROC | Ret. | Eng. | h-AUROC | Ret. | Eng. |
| 100% | .786 ± .033 | 100% | 85 | .853 ± .004 | 100% | 100 |
| 10% | .772 ± .059 | 98% | 9 | .830 ± .018 | 97% | 10 |
| 5% | .730 ± .018 | 93% | 4 | .709 ± .131 | 83% | 5 |
| **2%** | **.724 ± .013** | **92%** | **2** | .635 ± .065 | 74% | 2 |
| 1% | .670 ± .110 | 85% | 1 | .513 ± .220 | 60% | 1 |

## A. Dataset Overview

## B. Label Efficiency

## C. Baseline Comparison Protocol

All five methods share: (1) matched downstream capacity (horizon-conditioned MLP, trained with positive-weighted BCE), (2) identical evaluation (h-AUROC, same splits, same horizon sets), and (3) identical label budgets.

For HEPA, the finetuned component is the pretrained predictor MLP (197.6K params) plus a shared linear event head (769 params), totalling 198K. For all baselines, we use a dt-conditioned MLP head: linear projection to 256, learned horizon embedding, LayerNorm, and a 3-layer MLP ($256 \rightarrow 256 \rightarrow 256 \rightarrow 1$), totalling 264K parameters—giving baselines slightly *more* downstream capacity than HEPA.

The methods differ only in how the frozen encoder is obtained:

- **PatchTST** (Nie et al., 2023): channel-independent patching, supervised end-to-end, 5 seeds.
- **iTransformer** (Liu et al., 2024): variate-attention, supervised end-to-end, 5 seeds.
- **MAE**: same architecture as HEPA, masked reconstruction objective (isolates JEPA vs. MAE under identical architecture), 5 seeds.
- **Chronos-2** (Ansari et al., 2025): 119M-param foundation model, per-channel extraction + mean pooling, 3

seeds.

By freezing all encoders, differences in h-AUROC reflect encoder quality alone.

## D. Finetuning-Mode Ablation

*Table 4.* **Finetuning-mode ablation on C-MAPSS (5 seeds, F1w; earlier architecture variant with EMA target).** Pred-FT outperforms scratch by +0.226 at 5% labels ($p=0.039$, $d=1.35$). Under the final architecture (198K pred-FT, SIGReg), pred-FT and E2E achieve equivalent performance ($\Delta \leq 0.003$).

| | 100% labels | | 5% labels | |
|---|---|---|---|---|
| Mode | F1w ↑ | RMSE ↓ | F1w ↑ | RMSE ↓ |
| probe_h (257 p) | 0.30 ± 0.06 | 16.0 ± 1.5 | 0.06 ± 0.10 | 20.4 ± 1.2 |
| frozen_multi (4K p) | 0.15 ± 0.03 | 19.0 ± 0.1 | 0.18 ± 0.14 | 24.4 ± 4.9 |
| **pred_ft (790K p)** | 0.39 ± 0.09 | 16.9 ± 1.7 | **0.26 ± 0.17** | 24.3 ± 6.8 |
| e2e (2.37M p) | 0.41 ± 0.12 | 15.0 ± 1.2 | 0.18 ± 0.24 | **20.1 ± 1.9** |
| scratch (2.37M p) | 0.40 ± 0.08 | 14.5 ± 0.7 | 0.04 ± 0.05 | 32.9 ± 2.0 |

## E. Foundation Model Comparisons

Table 5 consolidates the matched-head comparison between HEPA (5 seeds) and four foundation models: Chronos-2 (119M), MOMENT-1-large (Goswami et al., 2024) (341M), TFM-2.5 (Das et al., 2024) (204M), and Moirai (Woo et al., 2024) (91M). All encoders are frozen; only the 198K-param downstream head is trained.

*Table 5.* **HEPA vs. four foundation models (matched head, 100% labels).** HEPA: 5 seeds; FMs: 3 seeds. **Bold** = best. "—": not run.

| Dataset | HEPA | Chr-2 | MOMENT | TFM-2.5 | Moirai |
|---|---|---|---|---|---|
| FD001 | **.73**±.02 | .66±.00 | .56±.01 | .53±.00 | .61±.00 |
| FD002 | **.58**±.01 | .50±.01 | .60±.00 | .60±.01 | .66±.00 |
| FD003 | **.82**±.02 | .72±.02 | .47±.01 | .62±.01 | .70±.00 |
| SMAP | **.60**±.03 | .53±.01 | — | .51±.03 | — |
| PSM | .55±.02 | .49±.00 | — | **.57**±.01 | .53±.01 |
| MBA | .75±.01 | .55±.01 | **.79**±.01 | .76±.01 | .57±.02 |
| GECCO | .81±.07 | .81±.01 | — | **.93**±.01 | .82±.01 |
| BATADAL | .64±.02 | .58±.01 | .54±.07 | **.65**±.01 | .36±.01 |
| ETTm1 | **.87**±.00 | .78±.01 | — | .59±.01 | .60±.00 |

## F. Theory: Event-Information Retention

Predictor finetuning rests on a premise: the pretrained encoder retains enough event-relevant information that a small downstream head can extract it. Let $H_t = f_\theta(X_{\leq t})$, $H^* = \bar{f}_\theta(X_{(t, t+\Delta t]})$, $\hat{H} = g_\phi(H_t, \Delta t)$, and $\eta(h) := P(E_{t+\Delta t}=1 \mid H^*=h)$.

**Proposition 1** (Event-Information Retention). *Suppose* (A1) $E_{t+\Delta t} \perp X_{\leq t} \mid H^*$, (A2) $\mathbb{E}[\|\hat{H} - H^*\|_2^2] \leq \varepsilon$, (A3) $\eta$ *is* $L$-*Lipschitz,* (A4) $\eta(H^*) \in [\underline{\eta}, \overline{\eta}] \subset (0,1)$ *a.s. Then* $I(H_t; E_{t+\Delta t}) \geq I(H^*; E_{t+\Delta t}) - C_\eta L^2 \varepsilon$, *where* $C_\eta = (2\underline{\eta}(1-\overline{\eta}))^{-1}$.

The proof proceeds in three steps. First, because $\hat{H}$ is a

deterministic function of $H_t$, the data processing inequality gives $I(H_t; E) \geq I(\hat{H}; E)$. Second, a Jensen-gap argument on convex KL divergence, combined with the Lipschitz condition and prediction error bound, yields $I(H^*; E) - I(\hat{H}; E) \leq C_\eta L^2 \varepsilon$. Combining these produces the result.

**Corollary 2** (Precursor necessity). *The bound is non-vacuous if and only if the future interval contains event precursors that the target encoder captures ($I(H^*; E_{t+\Delta t}) > 0$) and the predictor approximates the target well enough ($\varepsilon < I(H^*; E_{t+\Delta t})/(C_\eta L^2)$).*

This corollary explains both successes and failures. On C-MAPSS, degradation unfolds over hundreds of cycles, so $I(H^*; E_{t+\Delta t})$ is large and pretraining drives $\varepsilon$ small, yielding h-AUROC $\geq 0.81$. On datasets without extended precursors, the bound is vacuous regardless of pretraining quality. Full proof and assumption-by-assumption failure modes are in the extended version.

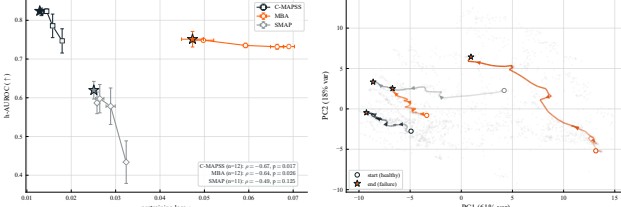

*Figure 2.* **Self-supervised pretraining learns task-relevant structure.** (a) Pretraining loss $\varepsilon$ vs. downstream h-AUROC at fixed checkpoints across three domains (3 seeds, error bars $\pm 1$ std). ★ marks the converged-best snapshot. (b) PCA of pretrained C-MAPSS-1 representations. PC1 captures 61% of variance; the encoder discovers a degradation manifold without any labels.

## G. MTS-JEPA Comparison

Table 6 compares HEPA against MTS-JEPA (He et al., 2026) under matched protocol (same context length, patch size, downstream head). The only difference is the SSL objective. HEPA wins on 8 of 9 datasets; MTS-JEPA wins on MBA.

*Table 6.* **HEPA vs. MTS-JEPA (matched head, h-AUROC).** HEPA: 5 seeds; MTS-JEPA: 1–3 seeds. TEP excluded (MTS-JEPA fails to converge on 52-dim input).

| Dataset | HEPA | MTS-JEPA |
|---------|------|----------|
| FD001 | **.81**$\pm$.04 | .69$\pm$.02 |
| FD002 | **.57**$\pm$.01 | .53$\pm$.02 |
| FD003 | **.84**$\pm$.02 | .78$\pm$.00 |
| SMAP | **.59**$\pm$.06 | .49$\pm$.00 |
| PSM | **.57**$\pm$.02 | .48$\pm$.00 |
| MBA | .75$\pm$.04 | **.88**$\pm$.00 |
| GECCO | **.88**$\pm$.06 | .84$\pm$.00 |
| ETTm1 | **.81**$\pm$.01 | .75$\pm$.00 |
| BATADAL | **.57**$\pm$.04 | .43$\pm$.00 |
| *HEPA wins* | | 8 of 9 |

