# OpenReview forum: "HEPA: A Self-Supervised Horizon-Conditioned Event Predictive Architecture for Time Series"
_ICML.cc/2026/Workshop/FMSD — FMSD @ ICML 2026 SpotlightOral_

### Official Review · Reviewer_3gFW · 2026-05-21
**Strong and well-executed JEPA-based framework for event prediction in multivariate time series**

**Rating:** 8
**Confidence:** 4

**Review:**

Review

This paper introduces HEPA, a self-supervised JEPA-style framework for multivariate time-series event prediction. The method pretrains a causal Transformer encoder using horizon-conditioned future representation prediction and later freezes the encoder while finetuning only the predictor and event head to produce a monotonic survival CDF over future horizons. The approach is evaluated on 14 benchmarks spanning 11 domains, including prognostics, anomaly detection, finance, and healthcare.

Overall, I found the paper technically strong, clearly written, and highly relevant to the workshop. The central idea of “predictor finetuning” after JEPA pretraining is interesting and empirically well-supported.

Strengths

* The paper is very well-motivated and presents a clean unifying formulation for event prediction across domains.
* The proposed predictor-finetuning setup is simple yet effective, achieving strong label efficiency while tuning substantially fewer parameters than end-to-end baselines.
* Experimental coverage is unusually broad for a workshop paper (14 datasets across 11 domains).
* The comparisons are generally fair: matched downstream heads, fixed protocols, and frozen encoders isolate representation quality well.
* The paper includes thoughtful analysis beyond benchmark tables, including:
    * label efficiency,
    * failure modes,
    * theory motivation,
    * MTS-JEPA comparison,
    * and pretraining-loss correlation analysis.
* The survival-CDF formulation is elegant and guarantees monotonicity by construction.

Weaknesses

* While the empirical study is strong, the methodological novelty is somewhat incremental, combining existing JEPA-style pretraining with survival modeling and lightweight finetuning.
* Theoretical claims are interesting but somewhat preliminary; the assumptions behind the information-retention proposition may be difficult to verify in practice.
* Several competing methods (especially foundation models) are evaluated only as frozen encoders. Some readers may question whether this fully reflects their capabilities.
* The paper admits some important failure modes (e.g., sensor-localized anomalies such as MBA), suggesting the architecture may still depend heavily on the tokenization strategy.

Comments

The paper is generally very polished and easy to follow. Figure 1 clearly communicates the two-stage training pipeline. I also appreciated the discussion of when the JEPA objective succeeds versus when the bound becomes vacuous. The breadth of experiments is a major strength.

I would have liked slightly more discussion about computational cost and pretraining stability, especially compared with large pretrained time-series foundation models. In addition, some benchmark improvements are relatively modest despite the strong framing around universality.

Overall, this is a solid workshop paper with strong experimental execution and a compelling representation-learning perspective for event prediction.

---

### Official Review · Reviewer_PRPi · 2026-05-21
**Interesting Two-Stage Event Prediction Framework Combining JEPA Pretraining and Survival CDF Modeling**

**Rating:** 8
**Confidence:** 4

**Review:**

## **Summary of contributions**

The paper proposes HEPA: A Self-Supervised Horizon-Conditioned Event Predictive Architecture for Time Series.
The method uses a two-stage training procedure: 1) JEPA-based pretraining of an encoder-predictor on unlabeled data to predict future latent representations (to avoid penalizing the model for unpredictable events), and 2) finetuning the predictor as a lightweight event prediction head on rare critical events using BCE losses over multiple future horizons. The approach outputs a discrete-time survival CDF.
Experiments show strong performance, achieving the best h-AUROC results on 10 out of 14 datasets. Comparisons are standardized by using the same event prediction head across 4 comparisons methods and varying only the encoder pretraining strategy.

## **Strengths and weaknesses**

### **Strengths**

- Interesting use of Joint Embedding Predictive Architecture, two-stage training framework for time series event prediction.
- The paper is well written and addresses a topic of interest to the community. The proposed method is clearly explained.

### **Weaknesses**

- Potentially limited comparison with methods specifically designed for survival/event prediction and anomaly detection.
## **Suggestions**

- It could be interesting to compare additional baselines using the domain-specific metrics, and additional information about the state of the art for each domain-specific task.

---

### Official Review · Reviewer_AHvr · 2026-05-22
**HEPA, a timeseries and CDF translation of JEPA**

**Rating:** 8
**Confidence:** 3

**Review:**

This paper introduces HEPA (Horizon-conditioned Event Predictive Architecture), a self-supervised framework designed to predict critical events in multivariate time series using minimal labeled data combining forecasted conditional hazards into a CDF. HEPA shifts the forecasting focus from noisy, raw sensor values to predictable latent dynamics (with JEPA).

Strengths:

- The paper is technically sound, well-structured, and explicitly addresses real-world constraints like data scarcity and parameter limits.
- Creative translation of LeCun's JEPA paradigm from visual frames to the time-series domain. It rethinks self-supervised time-series token processing by choosing to forecast representations rather than values.
- Guaranteed monotonicity
- Exceptional parameter efficiency. The model achieves state-of-the-art or highly competitive performance using an encoder footprint of just ~1.96M parameters, proving that multi-horizon event prediction does not inherently require the 100M+ parameter scale of the benchmarked foundation models.
- Well placed in related work

Areas for Improvement:

- Foundational models are not finetuned, which induces an unfair comparison between a fully aligned/pretrained model on the data and the frozen foundational models. These models are natively optimized for generative value-forecasting rather than horizon-conditioned event tracking.
- Have you experimented with pretraining a single unified encoder across a collection of datasets with a matched number of channels (this regards the workshops wording: ‘challenging the traditional 'predictive foundation models... train per dataset' paradigm’?)
- Comprehensive computation speeds (such as inference latency, execution throughput, or downstream finetuning times) are not computed or benchmarked.
- Encoder model size ablation could be insightful.

Justification of Score:

Despite limitations regarding frozen baselines and a 'train-per-dataset' paradigm, this original, parameter-efficient application of JEPA justifies acceptance. The architectural recipe and monotonic CDF yield strong empirical results, though future iterations must include compute metrics and fine-tuned baselines.